# In Vitro Anti-Tumor and Hypoglycemic Effects of Total Flavonoids from Willow Buds

**DOI:** 10.3390/molecules28227557

**Published:** 2023-11-13

**Authors:** Peng Zhang, Lulu Fan, Dongyan Zhang, Zehui Zhang, Weili Wang

**Affiliations:** 1College of Life Engineering, Shenyang Institute of Technology, Fushun 113122, China; fll135126@163.com (L.F.); z182313@163.com (D.Z.); 2College of Laboratory Animal Medicine and Science, Liaoning University of Traditional Chinese Medicine, Shenyang 110847, China; zhangzehuisyau@163.com; 3Liao Ning Institute for Drug Control, Shenyang 110031, China

**Keywords:** willow buds, total flavonoids, Q-Orbitrap LC-MS/MS, anti-tumor, hypoglycemic

## Abstract

*Salix babylonica* L. is a species of willow tree that is widely cultivated worldwide as an ornamental plant, but its medicinal resources have not yet been reasonably developed or utilized. Herein, we extracted and purified the total flavonoids from willow buds (PTFW) for component analysis in order to evaluate their in vitro anti-tumor and hypoglycemic activities. Through Q-Orbitrap LC-MS/MS analysis, a total of 10 flavonoid compounds were identified (including flavones, flavan-3-ols, and flavonols). The inhibitory effects of PTFW on the proliferation of cervical cancer HeLa cells, colon cancer HT-29 cells, and breast cancer MCF7 cells were evaluated using an MTT assay. Moreover, the hypoglycemic activity of PTFW was determined by investigating the inhibitory effects of PTFW on α-amylase and α-glucosidase. The results indicated that PTFW significantly suppressed the proliferation of HeLa cells, HT-29 cells, and MCF7 cells, with IC50 values of 1.432, 0.3476, and 2.297 mg/mL, respectively. PTFW, at different concentrations, had certain inhibitory effects on α-amylase and α-glucosidase, with IC50 values of 2.94 mg/mL and 1.87 mg/mL, respectively. In conclusion, PTFW at different doses exhibits anti-proliferation effects on all three types of cancer cells, particularly on HT-29 cells, and also shows significant hypoglycemic effects. Willow buds have the potential to be used in functional food and pharmaceutical industries.

## 1. Introduction

Cancer is emerging as a major public health challenge globally, with high morbidity and mortality. The current medical treatments in clinical practice mainly include synthetic drugs, but these are always accompanied by significant side effects and high costs. Since the 1980s, the successful application of plant-derived anticancer drugs, represented by Paclitaxel, has prompted extensive research on plant-derived anticancer drugs [1]. Plant-derived drugs used for anticancer treatment can protect healthy cells while targeting cancer cells. The combination of plant extracts and chemotherapy drugs can amplify anticancer effects by regulating various signaling pathways, such as P13K/AKT, NF-κB, JNK, ERK, and WNT/β-catenin [2].

In recent years, diabetes mellitus has become the third most common chronic non-infectious disease after cardio-cerebrovascular diseases and tumors. The incidence of type 2 diabetes mellitus (T2DM), accounting for more than 90% of all cases of diabetes, is rising year by year [3,4]. The chemical drugs used to treat hyperglycemia, such as acarbose and voglibose, have varying degrees of adverse reactions. By contrast, plant-derived extracts are ideal hypoglycemic and lipid-lowering drugs by virtue of their abundant active ingredients. Hence, the search for natural anti-diabetic drugs has become a popular topic in the field of diabetes treatment.

The roots, branches, bark, leaves, and flowers of *Salix babylonica* L. have traditionally been used as medicinal materials. Modern pharmacological studies have shown that *Salix babylonica* L. possesses a wide range of pharmacological activities, such as anti-inflammatory, analgesic, and antibacterial effects [5], as well as significant relieving effects on rheumatic pain, toothache, and burns. Moreover, *Salix babylonica* L. has notable efficacy in terms of eliminating edema and jaundice. As a medicine food homology plant, willow bud contains a variety of trace elements, such as protein, carrots, vitamins, carbohydrates, vitamin C, calcium, iron, and iodine, in addition to abundant flavonoids. Therefore, it has important application value in the medical, food, and beauty industries. This study aims to investigate the in vitro anti-tumor and hypoglycemic effects of purified total flavonoids from willow buds (PTFW), thereby providing a reference for the further development of Salix plants.

## 2. Results and Discussion

### 2.1. PTFW Content

The extraction rate of PTFW was calculated, using rutin as the standard. The flavonoid content range of was 0–10 μg/mL, and showed a good linear relationship within this range. According to the regression equation y = 0.1119x − 0.0004 (R^2^ = 0.9995), the total flavonoid content in the willow buds was calculated to be 80%.

### 2.2. Chemical Component Analysis of PTFW Extract

The chemical components of PTFW were obtained by means of liquid chromatography–mass spectrometry (LC-MS), and were characterized in different databases (including mzCloud, mzVault, and ChemSpider). In addition, the compounds with peak areas greater than 1 × 10^8^ were analyzed in order to screen out valuable components for future research. The Rt, [M − H]^−^, MS/MS [M − H]^−^, [M + H]^−^, MS/MS [M + H]^−^, calculated mass, and formula of each component are listed in Table 1. The distributions of primary and secondary mass spectra of each component are shown in Figure 1. The fragment ions of compounds 1, 2, 5, 7, and 10 were compared with our previously reported data, and *m*/*z* 289.1 and 203.1, 245.1 were considered to be catechin [3]; *m*/*z* 287.1 and 609.1 were considered to be kaempferol and rutin [5]; and *m*/*z* 303.04 and 317.06 ion peaks were consistent with quercetin and isorhamnetin peaks. Moreover, the ion peaks of *m*/*z* 464.38 and 448.37 were similar to those of myricitrin and cynaroside, which we also screened [6]. Through a comparison between various studies in the literature, *m*/*z* 463.0879, 431.0, and 447.0933 were found to match isoquercitrin [7], apigetrin [8], and apigenin 7-*O*-glucuronide [9].

### 2.3. Inhibitory Effect of PTFW on Three Types of Cancer Cells

Flavonoids are widely distributed in plants, fruits, and grains, and can be used as important candidate drugs for cancer prevention. Flavonoids, such as quercetin, kaempferol, apigenin, myricetin, baicalein, luteolin, and isorhamnetin, have shown promising anticancer effects [10,11,12,13]. The anticancer effects of plant extracts have also been extensively researched. For example, 500 μg/mL of lotus-leaf-enriched flavonoid extracts can inhibit the proliferation of colon cancer HT-29 cells [14]. Aqueous fenugreek seed extract can effectively fight against pancreatic cancer [15], and silymarin shows potential therapeutic efficacy in gastrointestinal malignancies [16]. Our experimental results found that treatment with different doses of PTFW inhibited the proliferation of cervical cancer HeLa cells, colon cancer HT-29 cells, and breast cancer MCF7 cells in a dose-dependent manner (Figure 2), with IC50 values of 1.432, 0.3476, and 2.297 mg/mL, respectively. Particularly, we found that PTFW exhibited the optimal inhibitory effect on colon cancer HT-29 cells. In addition, we observed that, under a microscope, the Hela cells in the drug group had larger morphological gaps, gradually disappeared connections, decreased numbers, partial floating cells in the culture medium, and significantly reduced proliferation activity compared with the blank control group. The MCF7 cells in the drug group became smaller were irregularly shaped, appearing in long spindles or even pointed shapes. The gap between adjacent cells within the cluster increased, as did the gap between adjacent cells. The HT29 cells in the drug group showed larger gaps, smaller volumes, and rounder shapes (Figure 3).

After 72 h of treatment with different doses of PTFW (0.25, 0.5, 1, 2, and 4 mg/mL), cell proliferation was detected using the MTT assay. The data are expressed as the mean ± standard deviation obtained from three independent measurements.

### 2.4. In Vitro Hypoglycemic Test

α-amylase and α-glucosidase are key enzymes affecting the digestion and absorption of major carbohydrates, such as starch and sugars. Their inhibitors have commonly been used in the treatment of T2DM, as they can competitively inhibit the activity of enzymes, delay the absorption of intestinal carbohydrates, and thus effectively suppress the rise of postprandial blood glucose [17]. Therefore, research concerning efficient and safe α-amylase and α-glucosidase inhibitors from natural products has become popular in the development of anti-diabetic drugs. As shown in Figure 4, PTFW inhibited α-amylase (Figure 4A) and α-glucosidase (Figure 4B) in a dose-dependent manner. The IC50 value results of different samples of α-amylase and α-glucosidase are shown in Table 2. As shown in Table 2, although the inhibitory effect of PTFW was found not to be as robust as that of acarbose, it still has application prospects as a hypoglycemic supplement. Plant-derived extracts have commonly been used as hypoglycemic drugs. For instance, mulberry leaves have been used for thousands of years as traditional hypoglycemic medicinal plants in China. Modern research has found that the alkaloids, flavonoids, and polysaccharides in mulberry leaves exert hypoglycemic activity by regulating glucose, amino acids, and lipid metabolism [18]. Similarly, the hypoglycemic effects of *Acacia* [19] and *Salvia japonica* [20] have been confirmed in previous studies.

## 3. Materials and Methods

### 3.1. Plant Materials

The willow buds of *Salix babylonica* L. (~1 cm) were collected at the Shenyang Institute of Technology from March 2022 to April 2022 and authenticated by Prof. Junfan Fu from the College of Life Engineering, Shenyang Institute of Technology. After washing, the collected willow buds were naturally dried and ground into fine powders for further experiments.

### 3.2. Preparation of PTFW

The dried willow bud powders were subjected to ultrasound extraction for 35 min at a liquid-to-material ratio of 70:1, with an ethanol concentration of 50%, followed by concentration and rotary evaporation to obtain the crude extract (CTFW) [5]. The filtrate was adsorbed by pre-treated D101 resin, with the optimal parameters of the sample solution being a pH of 3, a mass concentration of 0.8 mg·mL^−1^, and a volume of 2 BV. The CTFW was processed and concentrated in a rotary evaporation device, followed by vacuum drying for further analysis.

### 3.3. Determination of PTFW Content

The content of flavonoids was determined by aluminum nitrate colorimetry, with rutin as the standard [21]. The standard solution was prepared using rutin solution. Briefly, 2 mL of the extract was mixed with NaNO_2_ (0.8 mL, 5%), allowed to stand for 6 min, and then added to AlCl_3_ (0.8 mL, 10%) and NaOH (10 mL, 1 mol/L). Finally, the solution was diluted to 25 mL with 30% ethanol. After 15 min, the content of TFW was measured at the wavelength of 510 nm: y = 0.1119x − 0.0004 (R^2^ = 0.9995), y: absorbance, x: content. The purity of TFW was calculated using Formula (1).
Content (%) = (C × V)/M1 × 100%(1)
where C is the total flavonoid concentration in the sample solution (mg·mL^−1^), V is the volume of the sample solution (mL), and M1 is the dry powder mass after drying (mg).

### 3.4. Component Analysis and Detection Conditions

In reference to the previously reported methods [5], 150 mg of the total flavonoid powder of the willow bud (PTFW) was accurately weighed, then dissolved in 1 mL of 80% methanol solution and ground for 5 min. This was followed by vortexing for 10 min and centrifugation at 4 °C and 20,000× *g* for 10 min. The supernatants were filtered with a 0.22 μm membrane prior to ultra-performance liquid chromatography (UPLC) and LC-MS analysis.

Ultra-performance liquid chromatography (UPLC) was performed using an AQ-C18 column (150 × 2.1 mm, 1.8 μm; Welch, Concord, MA, USA). The column was maintained at 35 °C and evaporated at a flow rate of 0.30 mL/min. The mobile phase consisted of water with 0.1% formic acid (A) and acetonitrile with 0.1% formic acid (B). The elution details are shown in Table 3. Needle wash: methanol; autosampler temperature: 10 °C; autosampler syringe height: 2.00 mm; automatic injection volume: 5.00 μL.

Liquid chromatography–mass spectrometry (LC-MS) was carried out under a positive/negative electrospray ionization source (ESI) switching pattern. The condition parameters were as follows: detection method: full mass/dd-MS2; resolution: 70,000 (full mass); 17,500 (dd-MS2); scan range: 100.0–1500.0 *m*/*z*; spray voltage: 3.2 kV (positive); data acquisition time: 30 min; collision gas: high-purity argon (≥99.999%); sheath gas: nitrogen (≥99.999%); auxiliary gas: nitrogen (≥99.999%); 350 °C. The capillary temperature was kept at 300 °C.

### 3.5. Cell Culture

The cervical cancer HeLa cells, colon cancer HT-29 cells, and breast cancer MCF7 cells were purchased from Shanghai Solarbio Biotechnology Co., Ltd. (Shanghai, China). The HeLa and MCF7 cells were cultured in Dulbecco’s modified eagle medium (DMEM) containing 10% fetal bovine serum (FBS) and 50 U/mL penicillin/streptomycin, while HT-29 cells were cultured in RPMI-1640 medium containing 10% FBS and 50 U/mL penicillin/streptomycin, in a humidified incubator with 5% CO_2_ at 37 °C.

### 3.6. TFW Inhibited Cancer Cell Proliferation

The cell proliferation was measured using a 3-(4,5-dimethylthiazol-2-yl)-2,5-diphenyltetrazolium bromide (MTT) assay [22]. The cells were seeded into 96-well plates (100 μL/well) at a concentration of 1 × 10^5^ cells/mL. After cell attachment, the experimental groups were added to 100 μL of test drug culture medium at different concentrations for final concentrations of 0.5, 1, 2, and 4 mg/mL. The positive control group was added to 5-fluorouracil(5-FU) at a final concentration of 4 μg/mL, while the negative control group was added to the culture medium containing 0.5% dimethyl sulfoxide (DMSO) (100 μL/well). The blank control group was added to the same volume of culture medium, with 4 parallel wells in each group. After incubation at 37 °C for 72 h, the cells were supplemented with 20 μL of MTT solution (5 mg/mL) for another 4 h of incubation. Then, the supernatant was discarded, and 150 μL of DMSO was added to each well and mixed thoroughly. The absorbance (A570) was measured using a microplate reader (λ = 570 nm). The inhibition rate of the drugs on tumor cell growth was calculated according to the following formula: cancer cell growth inhibition rate (%) = [1 − A570 (experimental group)/A570 (control group)] × 100%.

### 3.7. In Vitro Hypoglycemic Test

For this step, 3,5-dinitrosalicylic acid colorimetry (DNS assay) was adopted [8]. The TFW was prepared in solutions with mass concentrations of 1, 2, 4, 8, and 16 mg/mL, respectively. The samples were dissolved in phosphate buffer solution (PBS, 0.067 mol/L, pH 6.8) before use. Then, 0.3 mL of α-amylase solution (0.686 mg/mL) was mixed with the sample evenly, preheated in a 37 °C water bath for 5 min, and added to 0.3 mL of 1% soluble starch that had been preheated at 37 °C for 5 min. After mixing evenly and reacting for 15 min, 0.5 mL of DNS chromogenic agent was added immediately for color development, and the reaction was terminated. The ultraviolet spectrophotometer was used to measure the absorbance value (A) of the solution at 540 nm, and its inhibition rate on α-amylase was calculated according to Formula (2):Inhibition rate (%) = [1 − (A1 − A2)/(A3 − A4)] × 100% (2)
where A1 is the absorbance value measured by the sample group; A2 is the absorbance value measured by replacing α-amylase with an equal volume of PBS under the same conditions; A3 is the absorbance value measured by replacing the sample with an equal volume of PBS under the same conditions; and A4 is the absorbance value measured by replacing the sample and α-amylase with an equal volume of PBS under the same conditions. The experiments were independently repeated 3 times.

The 4-nitrophenyl-β-D-glucopyranoside (PNPG) method was used [23]. The TFW was dissolved in phosphate buffer solution (PBS, pH 6.8) to obtain solutions with mass concentrations of 0.025, 0.05, 0.1, 0.2, and 0.4 mg/mL, respectively. Then, 20 μL of the sample solution and 20 μL of 2 U/mL α-glucosidase solution were incubated in 96-well plates at 37 °C for 15 min and added to 20 μL of 25 mmol/L PNPG solution (the solvent was 0.1 mol/L PBS) for 10 min of incubation at 37 °C. Finally, 100 μL of 0.2 mol/L Na_2_CO_3_ solution was added to terminate the reaction. The absorbance value (A) at the 405 nm wavelength was measured using a microplate reader, and its inhibition rate on α-glucosidase was calculated according to Formula (3):Inhibition rate (%) = [1 − (A2 − A1)/A0] × 100% (3)
where A0 is the absorbance value measured by replacing the sample with distilled water under the same conditions; A1 is the absorbance value measured by replacing α-glucosidase with distilled water under the same conditions; and A2 is the absorbance value measured by the sample group. The experiments were independently repeated 3 times.

### 3.8. Statistical Analysis

SPSS17.0 software was adopted for statistical analysis. All data were expressed as the mean ± standard deviation. GraphPad Prism was used to draw the line chart and calculate the inhibitory concentration (IC50).

## 4. Conclusions

In conclusion, we identified 10 kinds of flavonoid compounds through Q-Orbitrap LC-MS/MS technology, including flavones (apigetrin, cynaroside, apigenin 7-*O*-glucuronide), flavan-3-ols (catechin), and flavonols (isorhamnetin, myricitrin, kaempferol, quercetin, rutin, and isoquercitrin), and further proved the anticancer and hypoglycemic effects of the total flavonoids from willow buds. Despite the promising pharmacological effects of the total flavonoids from willow buds, their specific mechanism of action still requires further exploration. This study demonstrated the biological activity of purified total flavonoids from willow buds, providing a reference for the further development and utilization of willow buds with medicinal and edible homology.

## Figures and Tables

**Figure 1 molecules-28-07557-f001:**
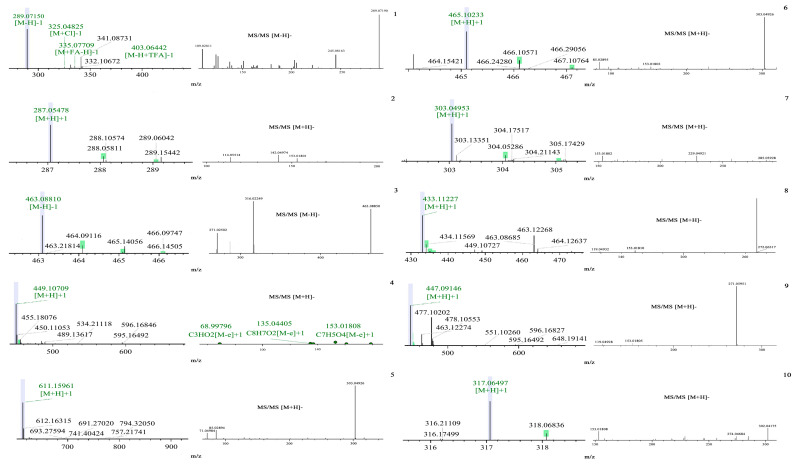
MS and MS/MS spectra profiles of 10 constituents of PTFW, measured by LC-MS.

**Figure 2 molecules-28-07557-f002:**
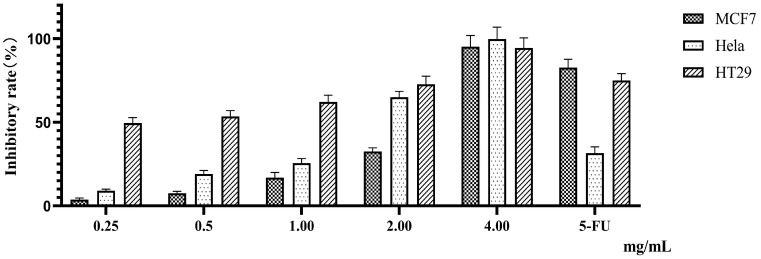
Antiproliferative effects of PTFW on cervical cancer HeLa cells, colon cancer HT-29 cells, and breast cancer MCF7 cells.

**Figure 3 molecules-28-07557-f003:**
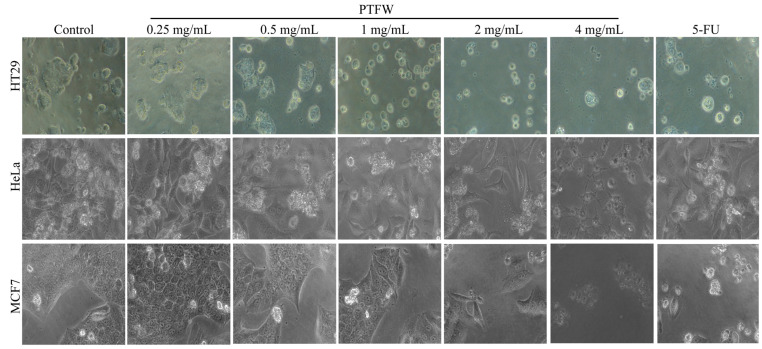
Effects of different concentrations of PTFW on the morphology of cervical cancer HeLa cells, colon cancer HT-29 cells, and breast cancer MCF7 cells.

**Figure 4 molecules-28-07557-f004:**
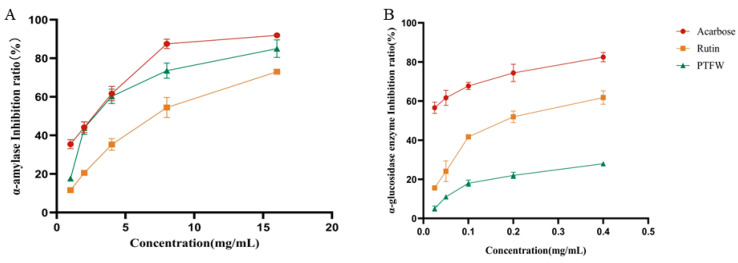
Inhibitory effects of PTFW on α-amylase and α-glucosidase activity.

**Table 1 molecules-28-07557-t001:** Identification of the chemical components of PTFW extract.

No.	Rt (min)	[M − H]^−^	MS/MS [M − H]^−^	[M + H]^−^	MS/MS [M + H]^−^	Calculated Mass	Formula	Proposed Molecule	References
1	9.00	289.071	109.028, 245.081, 289.071	—	—	290.1	C_15_H_14_O_6_	Catechin	[5]
2	12.38	—	—	287.055	114.055, 142.049, 153.018	286.1	C_15_H_10_O_6_	Kaempferol	[5]
3	12.59	463.088	271.025, 316.022, 463.088	—	—	464.1	C_21_H_20_O_12_	Myricitrin	[6]
4	12.63			449.107	68.997, 135.044, 153.018	448.1	C_21_H_20_O_11_	Cynaroside	[6]
5	12.81	—	—	611.159	71.049, 85.028, 303.049	610.2	C_27_H_30_O_16_	Rutin	[5]
6	12.86	—	—	465.102	85.028, 153.018, 303.049	464.1	C_21_H_20_O_12_	Isoquercitrin	[7]
7	12.87	—	—	303.049	153.018, 229.049, 285.123	302.0	C_15_H_10_O_7_	Quercetin	[5]
8	13.34	—	—	433.112	119.049, 153.018, 272.063	432.1	C_21_H_20_O_10_	Apigetrin	[8]
9	13.55	—	—	447.091	119.049, 153.018, 271.059	446.1	C_21_H_18_O_11_	Apigenin 7-*O*-glucuronide	[9]
10	13.63	—	—	317.064	153.018, 274.048, 302.041	316.1	C_16_H_12_O_7_	Isorhamnetin	[5]

**Table 2 molecules-28-07557-t002:** IC50 determination results of α-amylase and α-glucosidase in different samples.

Sample	IC50 mg/mL	
	α-Amylase	α-Glucosidase
Acarbose	2.124	0.01542
Rutin	6.730	0.1895
PTFW	2.943	1.874

**Table 3 molecules-28-07557-t003:** Chromatographic gradient elution.

Time (min)	Water Phase Ratio (%)	Organic Phase Ratio (%)
1	98	2
5	80	20
10	50	50
15	20	80
20	5	95
27	5	95
28	98	2
30	98	2

## Data Availability

The original contributions presented in the study are included in the article; further inquiries can be directed to the corresponding author.

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
