# Peer review of "In Vitro Anti-Tumor and Hypoglycemic Effects of Total Flavonoids from Willow Buds"

_molecules, 2023, doi:10.3390/molecules28227557_

Round 1

Reviewer 1 Report

Comments and Suggestions for Authors

This manuscript describes a preliminary biological evaluation of purified flavonoid extraction from willow buds (PTFW). In my opinion, this study is not suitable and does not address the criteria of the journal. Authors report only a characterization of an extraction and a preliminary biological evaluation by providing results from 2 only assays. As it concerns the antiproliferative effect studies, authors must run for comparison purposes the plain flavonoids in a purified form. In addition, the images in Figure 3 are not presentable a must be replaced with ones of better resolution.

Comments on the Quality of English Language

Reviewer 2 Report

Comments and Suggestions for Authors

The following points to be addressed and discussed in the article. 

1. TFW to be expanded as like PTFW.

2. From line no: 175.  The entire content need to be addressed appropriately. 

Approximately 0.15g of traditional Chinese medicine sample was weighed, added  with 1000µL of 80% methanol, and grounded with grinding media for 5 min, followed by  vortex for 10 min and centrifugation at 4℃ and 20000xg for 10 min. The supernatant was  filtered for analysis[5].  

Define the ?Chinese medicine.  Is it the extracted compound?  if not what and where is the role of Chinese medicine in this study?

Line no: 170 and 171.  There is a number for  the formula to calculate TWF  similarly may give the formula number to calculate inhibition rate on α-amylase in the line 228 and 244.

Recommended to provide possible high resolution MS/MS spectra figure for each compound with indicated peaks. 

Justify why the type of cell death (Apoptosis or Necrosis)  study not included in this study. 

Why not used the control normal cell line study to prove the efficacy of this study. 

Round 2

Reviewer 1 Report

Comments and Suggestions for Authors

The paper is suitable for publication, authors addressed all the raised comments

Author Response

Thank you very much for your recognition

Reviewer 2 Report

Comments and Suggestions for Authors

Appreciate the revision 

Author Response

Thank you very much for your review.